

# Spatial Signature of Solar Forcing over the North Atlantic Summer Climate in the Past Millennium

**Maria Pyrina[1],[*], Eduardo Moreno-Chamarro[2], Sebastian Wagner[1], Eduardo Zorita[1]**

[1]Helmholtz Zentrum Geesthacht, Institute of Coastal Research, Geesthacht, 21502, Germany

5  [2]Barcelona Supercomputing Center, Barcelona, 08034, Spain

[*]maria.pyrina@hzg.de

## Abstract

We investigate the effects of solar forcing during summer on the North Atlantic climate in comprehensive simulations of the preindustrial last millennium. We use two Earth System Models forced only by variations in Total Solar Irradiance (TSI). Specifically, we examine how different statistical techniques commonly used in current literature, namely linear methods and composite techniques can condition our understanding of the effects of solar forcing on climate. We demonstrate that the results obtained are strongly shaped by internal model variability. Linear methods like regression and correlation are not suitable to separate solar impacts on summer climate from internal variability. Composite maps show a response of SSTs off the European coasts and atmospheric blocking-like pressure anomalies over the subpolar North Atlantic, with some model-dependent variations of its spatial patterns and extent. In the models analyzed, the relationship of TSI to the tropospheric and surface circulation is linked through a baroclinic response to diabatic heating at the ocean surface. A tendency toward blocking-like patterns over the middle and high latitudes might be subsequently created during summer and in high TSI periods.

## 1 Introduction

Investigating the Sun-climate relationship prior to industrial times is important for the comparison between climate models and proxies and for disentangling anthropogenic climate change from natural variations. The Sun's climatic impact arises from changes in its total radiative output (the Total Solar Irradiance; TSI), spectral solar irradiance (SSI) and energetic particle precipitation (EPP) (Haigh, 1994;Lean, 1997;Seppälä and Clilverd, 2014;Friis-Christensen and Svensmark, 1997;Shindell et al., 1999;Lilensten et al., 2016). Recently, more efforts have been placed on the representation of variations in SSI and EPP in climate model simulations, for example, motivated by the interest in investigating the climate sensitivity to UV forcing (Ball et al., 2016;Ineson et al., 2015;Langematz et al., 2013;Maycock et al., 2018). Climate models included SSI variations already before the 5th phase of the Coupled Model Intercomparison Project, CMIP5 (Austin et al., 2008;Haigh, 1996;Matthes et al., 2006;Rozanov et al., 2004;Ermolli et al., 2013;Haigh et al., 2010). Nevertheless, there is still the need for a suitable resolution in climate model radiation schemes in order to capture solar-induced wavelength-dependent changes (Hood et al., 2015;Misios et al., 2016). Currently, there is a new recommendation regarding the solar forcing dataset for CMIP6 with higher SSI variations in comparison to the one recommended for CMIP5 (http://solarisheppa.geomar.de/cmip6). The SSI uncertainty and the possible impacts of the higher SSI variations are discussed extensively in the paper of Matthes et al., 2017.

In the current study, we are interested in the climatic impact of TSI variations. The effect of TSI changes on the climate of the past millennium was identified by early works in the context of the historical Maunder Minimum (AD 1675–1715) (Eddy et al., 1976) and confirmed by more recent



studies investigating climate changes during periods of solar maxima and minima (Ermolli et al., 2013;Haigh, 2007;Ineson et al., 2011;Chiodo et al., 2016;Ineson et al., 2015). In the early 2000s it was generally accepted that a long-term change in solar activity between the mean state of the sun during the Maunder Minimum (decline in solar activity) and the present-day climate, a period with presumably higher solar output, was about 0.3% (Fligge and Solanki, 2000). However, the amplitude of TSI variations was subsequently  revised and reduced to approximately 0.1% (Vieira et al., 2011;Wang et al., 2005), a scaling that was already applied to CMIP5 models (Kopp and Lean, 2011;Prša et al., 2016). Even though this change in the TSI scaling has implications for our understanding of the Sun-climate coupling, a quantitative assessment has not been made yet, especially regarding spatial climatic responses. Therefore, in our study we use simulations forced with the reduced long-term solar amplitude and investigate how TSI variations may affect the North Atlantic (NA) climate in the preindustrial last millennium.

We focus on the NA, where ocean-atmosphere interactions control climate variability and crucially affect the climate of North America and Europe (Enfield et al., 2001;Hurrell and Folland, 2002;Sutton and Dong, 2012). Several studies that investigate climate variability during winter showed that the North Atlantic Oscillation (NAO) and the Arctic Oscillation (AO) respond to solar forcing through a top-down mechanism (Ineson et al., 2011;Kodera and Kuroda, 2002;Shindell et al., 2001;Thiéblemont et al., 2015;Woollings et al., 2010;Ogi et al., 2003;Maycock et al., 2015). This mechanism can act together with a bottom-up mechanism and amplify the solar forcing at the surface (Dima and Voiculescu, 2016;Meehl et al., 2008;White et al., 1997;Meehl et al., 2009). However, the role of these mechanisms in driving Sun-climate interactions has not been explored in detail during summer.

In contrast to winter, a more direct response to solar forcing can be expected for the northern latitudes during the summer season, because of the daytime length, especially for latitudes north of 60° N. Moreover, the summer season is more relevant for studies using proxy data. High-resolution (annual) climate reconstructions are derived from trees and bivalve shells. The variability in their growth increment widths is related to environmental conditions and their growth is biased towards summer (Fritts, 2012;Schöne et al., 2005). Many studies that use proxy data in order to investigate the Sun-climate relationship compare their results to model simulations (Novello et al., 2016;Zambri and Robock, 2016;Reynolds et al., 2016). Modeling studies and studies based on proxy data disagree on the importance of TSI forcing on driving variations in global annual mean surface temperatures (Hegerl et al., 2003;Wang et al., 2019;Swingedouw et al., 2011), but they agree on larger regional effects of TSI forcing (Sejrup et al., 2010;Reynolds et al., 2016;Lean and Rind, 2008). Nevertheless, these effects might be too large to be attributed to direct solar heating, and positive feedbacks are required (Chen et al., 2015), particularly in regions that usually are cloud covered (Salby and Callaghan, 2006;van Loon et al., 2004;Sfîcă et al., 2018).

Our study focuses on the NA troposphere and the ocean surface during the summer and using the reduced TSI scaling investigates the questions: (1) how does solar forcing interact with summer surface climate and mean atmospheric circulation? and, (2) how is the solar signal transferred to the surface? The third question relates to the different methodologies which are used so far in literature for the investigation of the climatic impacts of TSI variability and which show a variety of results that might be method dependent. Therefore, the third question we seek to address in this study is: (3) how do different methodologies condition our understanding of the effects of solar forcing on climate?

One common method for the investigation of the Sun-climate relationship prior to industrial times is to analyze the mean difference between two periods in the past millennium with markedly





different solar activity, for instance, the late 17th century Maunder Minimum, a period of low solar activity (Crowley, 2000;Zorita et al., 2004), and the late 18th century, a period of relatively high solar activity. Modeling studies that applied this methodology have mostly used model simulations forced with the Lean TSI reconstruction (Lean et al., 1995), which exhibits changes of around 0.3% between the Maunder Minimum and the late 20th century. Using the Lean TSI reconstruction and a General Circulation Model (GCM) the global average temperature change between those periods was found approximately equal to 0.3°C (Shindell et al., 2001). This is similar to some reconstructions of the annual average North Hemisphere (NH) 1680-minus-1780 temperature change of about –0.2° ± 0.2° C (Mann et al., 1999).

An alternative methodology is composite analysis, also referred to as superposed epoch analysis or conditional sampling. This method is useful for isolating low amplitude signals within data in which background variability would otherwise obscure the signal detection (Laken and Čalogović, 2013). This is a non-parametric approach, meaning that the method is not solely based on parametrized families of probability distributions like i.e. Pearson correlation analysis. It takes advantage of a potentially spatially constrained fingerprint of the TSI responses. Studies that used this non-linear technique have obtained a globally averaged air surface warming of almost 0.2 °C during solar maxima relative to solar minima (Camp and Tung, 2007;Labitzke et al., 2002;Van Loon et al., 2007).

Regression analysis, even though is prone to yielding spurious results (Benestad and Schmidt, 2009;Ingram, 2007;Stott and Jones, 2009), has been the most commonly used methodology for investigating the climatic effects of solar forcing. Possible reasons for the occurrence of spurious results could be the non-Gaussian distribution of TSI variations, as linear-based regression techniques usually require Gaussian-distributed variables (Von Storch and Zwiers, 2001;Von Storch, 1999), the high temporal autocorrelation of the TSI record, or features in the TSI data that might be instrumental artifacts (Lockwood and Fröhlich, 2008;Scafetta and West, 2007). Moreover, nonlinear climate responses associated with atmosphere-ocean feedback processes can be falsely attributed using linear methods (Lockwood, 2012). Therefore, in order to check the consistency of results a null-hypothesis-test should be formulated and a proper methodology to falsify the null-hypothesis should be selected (Legras et al., 2010).

In the following study, we investigate the spatial climatic changes that are solely induced by changes in solar activity, during the summer seasons of AD 850–1850. For this goal, we use solar-only forced CMIP5-type simulations available for the Max Planck Institute Earth System Model (MPI-ESM) and the Community Earth System Model (CESM), over the North Atlantic. These simulations employ the reduced long-term solar amplitudes, smaller to that used in simulations carried out one or two decades ago. Solar activity changes are now in addition implemented for different wavelength bands. Both the MPI-ESM-P and the CESM models use a comparably high resolution of the stratosphere, prescribe estimated changes in the ozone concentrations depending on changes in solar activity, and are thus able to capture the proposed transfer mechanisms for solar–climate coupling due to changes in TSI and SSI (Giorgetta et al., 2013;Kinnison et al., 2007). The use of only-solar forced simulations is important for having a clearer solar forcing signal, as solar forcing effects might be concealed by the larger effect of volcanic eruptions, for example, in periods when TSI minima coincided with volcanic eruptions during the last millennium. In this case, actually, the use of linear regression might not be appropriate for detecting the signal of solar forcing (Lockwood, 2012). Moreover, the CESM output includes an ensemble of solar-only forced experiments. That is important in order to quantify the magnitude of model internal variability and the magnitude of the TSI effects on the modeled climate.



Other studies that also used simulations with reduced long-term solar amplitudes investigated
the recent historical period during winter and used fully forced experiments without considering
ensemble simulations (Le et al., 2016;Misios et al., 2016;Mitchell et al., 2015;Thiéblemont et al.,
2015). Nevertheless, model internal variability can significantly influence regional climate responses
during the last millennium (Otto-Bliesner et al., 2016). Our analysis focuses on summer season,
therefore being more relevant to model-proxy comparisons. Moreover, the use of only-solar forced
experiments during the preindustrial era, automatically excludes possible influences from volcanic
eruptions and anthropogenic forcing. Therefore, the investigation of the pure impact of changes in
solar activity on the simulated climate is more robust. Another aspect of the paper is that our analysis
includes different techniques for signal identification. Therefore, our results might help to disentangle
potential solar-only induced changes from those of the fully forced runs, focusing spatially on the
effect of model internal variability. This kind of analysis has not yet been carried out for the most
recent generation of Earth System Models (ESMs) using the reduced scaling of long-term solar
activity changes.

## 2 Data and Methods

### 2.1 Methods

We focus on the spatial climatic response pattern of the North Atlantic Ocean due to TSI
changes during summer, including changes in atmospheric circulation. The geographical domain is
between 80˚W–30˚E and 21˚N–75˚N. This analysis is separately performed on inter-annual time
scales and for decadal low-pass filtered series, using an eleven-year running mean filter. The
decadally filtered series were obtained by filtering the climatic parameters in the spectral domain (cut
off frequency=1/11) using a low-pass filtering technique (Schulzweida et al., 2012). The effects of
TSI on the summer climate (June-August) of the NA basin during the preindustrial era of the last
millennium (AD 850–1850) are analyzed using a fully forced (R1) and a solar-only forced (R4)
simulation of the MPI-ESM-P model. The fully forced simulation R1 is analyzed in order to
investigate the robustness of the results of the MPI-ESM model. Moreover, an ensemble of four only
solar forced simulations (E1, E2, E3 and E4) conducted with the CESM model was investigated for
the preindustrial period.

To identify the effects of TSI forcing on the NA climate, we used three methods: The first
method identifies the linear dependence between variables, using linear parametric techniques such
as Pearson Correlation and Linear Regression and takes into account the full preindustrial period, AD
850–1849. The second method compares two climatically different periods during the last millennium
traditionally associated with high and low solar activity periods (Eddy, 1976;Luterbacher, 2001). The
two periods we investigate in detail relate to the Medieval Climate Anomaly (MCA) and the Little
Ice Age (LIA). In the present study, we consider 100-year long periods within the MCA and LIA
epochs, spanning AD 1088–1187 for the MCA and AD 1500–1599 for the LIA. The selection of the
periods is based on an arbitrary choice of 100 years within the Spörer Minimum that has been linked
to cooling during the LIA epoch of the last millennium (AD 1450–1850), and a warm period that
could be possibly caused by high solar activity during the period of the MCA (AD 950–1250). The
third method used is Composite Analysis, focusing on the effect of prominent TSI periods with high
and low TSI values, respectively. Periods with high TSI are defined as those with positive TSI
anomalies exceeding the twofold standard deviation, whereas periods with low TSI are defined as
those with negative TSI anomalies exceeding in magnitude 1.4 times the standard deviation. These





asymmetric requirements were set in order to have approximately the same amount of selected cases
with positive and negative extreme TSI values, as the TSI distribution resembles a positively skewed
non-Gaussian distribution (Figure 1b). It must be noted that such analysis does not account for lagged
responses in the climate due to TSI changes. Nevertheless, in coupled systems involving stationary
oscillations, lags and phase relationships do not establish causality for the solar influence on climate
(Lockwood, 2012;Drijfhout et al., 1999;Moore et al., 2006).

A two-sided Student's t-test that accounts for the effect of serial correlation was applied on
the results of the linear methods. A mean difference two-sided test was also conducted to evaluate
the statistical significance of the results according to the methods MCA minus LIA and composite
analysis. Our null hypothesis H0 is that all of the variability can be explained by model internal
variability for the variables under consideration. If the null hypothesis is rejected at the alpha level
α=0.05, then our alternative hypothesis H1 stands, which is that the variability is forced by processes
different to internal variability. However, we should keep in mind that statistical significance does
not necessarily imply physical relevance. Especially for the linear methods, we use a large pool of
samples (n=1000) and as the sample size is important for the estimation of the nominal level of
statistical significance, very large samples sizes will ultimately lead to the phenomenon that even
very small physical quantities reach the statistical significance threshold for the alpha level under
consideration.

## 2.2 Model Simulations

We use two comprehensive Earth System Models (ESMs) participating in the 5th phase of the
CMIP project (https://pcmdi.llnl.gov/mips/cmip5/). In the simulations conducted with the MPI-ESM-
P model, the configuration of MPI-ESM for paleo-applications (Jungclaus et al., 2014), the
atmosphere has a T63/1.9° horizontal resolution and 47 hybrid sigma pressure levels (ECHAM
(Stevens et al., 2013)), and the ocean grid is curvilinear bipolar with an effective 1.5° resolution (near
the equator) and 40 z-levels (MPIOM (Jungclaus et al., 2013)). The L47 atmospheric grid extends to
0.01 hPa in the vertical domain, where the center of the topmost level in the upper mesosphere is set.
This vertical extension of the atmospheric grid includes for the first time a better resolved stratosphere
in CMIP simulations for MPI-M (Giorgetta et al., 2013). The solar spectrum is split into 14 spectral
bands and temporal variations of solar irradiance depend on the wavelength. Here the relative
variations among spectral bands are the result of a physically reasonable estimation, since the
information provided by the ice-core records cannot differentiate between spectral bands (Kinne et
al., 2013). The monthly average ozone concentrations for the period AD 850–1849 are also prescribed
and were calculated using the AD 1850–1860 monthly climatology of ozone concentrations from the
AC&C/SPARC Ozone Database as a basis. The ozone dependency on solar irradiance is represented
through regression coefficients between historical ozone concentrations and the annual 180.5 nm
solar flux (Jungclaus et al., 2014).

As ozone is a radiative active gas susceptible to shortwave radiation, its concentration varies
in concert with TSI changes. It is therefore important that this effect is included in model simulations
to represent solar–climate interactions better (Palmer et al., 2004). Variations of the incoming solar
UV radiation (120–350 nm) lead to changes in stratospheric ozone concentration and to heating that
amplify the effect of the initial UV radiation changes in the Earth's atmosphere (Gray et al., 2010).
In contrast, the visible and infrared bands, which are much more weakly absorbed in the atmosphere,
heat directly the Earth's surface and in turn the lower atmosphere. Therefore, the separation of solar
activity changes based on wavelength bands is important to be taken into account by climate models.
Moreover, larger UV forcing leads to a larger surface response to TSI changes (Ermolli et al., 2013).





The realization R1 of the MPI-ESM-P model is part of the past1000 runs and follows the PMIP3 protocol regarding the external forcings used. The Crowley and Unterman (2013) reconstruction is used for volcanic aerosol optical depth and effective radius, while the Pongratz et al., (2008) reconstruction is used to prescribe regionally resolved land use changes. For solar radiation the Vieira et al., (2011) total solar irradiance reconstruction over the Holocene is employed, with an increase in TSI of 0.1% from the 17th century Maunder Minimum to present time (Lohmann et al., 2015). The second simulation analyzed in this study is the R4 simulation of the MPI-ESM-P model, which is forced only by changes in TSI. This simulation is not a part of the CMIP5.

The second ESM is the Community Earth System Model-Last Millennium Ensemble (CESM-LME), which uses the Vieira et al., (2011) total solar irradiance reconstruction over the Holocene to produce a four member solar-only forced ensemble over the period AD 850–2005. The ocean component of CESM-LME has a 1° horizontal resolution, while the atmospheric component a 2° horizontal resolution with 66 vertical levels with an upper boundary at 140 km (Kinnison et al., 2007). The CESM model is stratosphere resolving and includes a fully interactive stratospheric chemistry module, calculating in-line photolysis rates for 66 bands (Chiodo and Polvani, 2016). For the calculation of both photolysis and heating rates the Solar Spectral Irradiance (SSI) forcing is used as an input, leading to a realistic response of the atmosphere to changes in solar forcing (Chiodo et al., 2012). Both the CESM-LME and the MPI-ESM-P simulations were analyzed for the preindustrial period AD 850–1850. For better comparison, the original model output was post-processed and re-gridded to a regular grid (1˚×1˚ degree horizontal resolution).

## 3 Results

Using the three methodologies described before, we investigate the mid-to-low tropospheric response to solar forcing over the North Atlantic. We assume that the simulated changes in solar forcing are integrated into the lower troposphere and therefore might affect directly and indirectly the surface climate. The analysis is performed for two different time scales (inter-annual and decadal), because the variability of the climatic variables can change with the time scale.

### 3.1 Linear Methods

The TSI time series is correlated with atmospheric and oceanic variables and the Pearson correlation coefficients are shown on inter-annual time scales in Figure 2. The first column of Figure 2 shows the correlation with sea surface temperatures (SSTs), Figure 3 the correlation with sea level pressure (SLP) and Figure 4 the correlation with geopotential height at 500 hPa (geo500). For the regression analysis, the TSI anomalies are normalized and regressed over the North Atlantic (NA) atmospheric and oceanic variables (second column of Figures 2, 3 and 4, respectively). The corresponding results for decadal time scales are shown in the Supplementary Information (SI), Figures S7-S10.

On inter-annual time scales, the correlation coefficient does not exceed values around $r \approx \pm 0.2$ for all atmospheric and oceanic variables and for both models. On decadal time scales, the TSI signal is generally weakly correlated ($r < \pm 0.3$) with SLP and SSTs in both models. Therefore, only a small fraction of the total variance of the lower tropospheric variable is statistically linked to changes in TSI. The inter-annual TSI regression patterns are consistent with the patterns on decadal time scales and show similarities to the correlation maps. In the CESM ensemble, TSI changes are connected with SST warming of maximum +0.15 K per one SD ($\pm 0.32$ W·m$^{-2}$) of TSI, but depending on the ensemble member, there is statistically significant warming ($p < 0.05$) over different regions of the NA





basin. Regarding the MPI-ESM model, statistically significant warming occurs over the Nordic Seas and regions of the subtropical gyre, while a cooling effect of -0.1 K is found over regions of the central NA basin and the west coast of Greenland.

In the MPI-ESM-P and on inter-annual time scales, TSI forcing is linked with an increase of the SLP and geo500 values in the central NA basin and in a decrease over Greenland (Figure 3, Figure 4). Yet, only the decrease of the SLP values over Greenland is statistically significant. Similar results are found in the ensemble members E1 and E3 of the CESM model. The results with the CESM ensemble suggest that internal climate variability obscures the climate response to TSI changes during
the preindustrial period of the last millennium. The regression patterns between TSI and the variables studied herein strongly depend on the individual ensemble member, with no statistically significant common characteristics found amongst ensemble members. This result cannot be confirmed with the single model realization performed with the MPI model, questioning the robustness of the conclusions drawn by a single model simulation.


### 3.2 Medieval Climate Anomaly versus Little Ice Age

The mean temporal difference, MCA (AD 1088–1187) minus LIA (AD 1500–1599), of the atmospheric and oceanic variables between those periods is shown for the MPI-ESM-P and the CESM ensemble in the third column of Figures 2, 3 and 4 for the SSTs, SLP, and geo500, respectively.
Statistically significant cooler SSTs during the MCA (~-0.6 K) are found northeast of Iceland in the MPI-ESM model and the E1, E2 and E4 realizations of the CESM model. According to the MPI-ESM model, regions east of Greenland, over the subpolar gyre, and the subtropics are warmer during the MCA, with maximum values of around +0.7 K. Statistically significant SST warming occurs over different regions of the NA during the MCA depending on the CCSM4 ensemble member under
consideration, signifying again the role of model internal variability in shaping climate changes between these two periods.

Statistically significant differences in SLP are found for the MPI-ESM over the subtropical regions (Figure 3). The CESM ensemble does not agree with such a response in SLP, with three out of four members showing a somehow similar response of low-pressure anomalies over Greenland,
albeit not statistically significant. For the MPI-ESM, the geo500 (Figure 4) is lower by approximately 10 gpm above Greenland and south of Nova Scotia, while it is higher by approximately 10 gpm in a V-shape area covering the mid and high latitudes. According to CESM, the general response of the geo500 level to the change of TSI forcing from the MCA to the LIA epoch is lower in comparison to the response of the respective parameters calculated using the MPI-ESM model, with the maximum
values reaching approximately ±8 gpm in the E3 and E4 ensemble members. The spatial response of the geopotential height is generally different among the CESM ensemble members and shows a different pattern in comparison to the MPI-ESM model.

The differences between the two models are apparent in the response of all parameters and methods studied so far, indicating the relevance of investigating the effect of TSI forcing using
different methods and different models. Internal variability plays a significant role and therefore the application of a method that will enhance the detection of the TSI signal by increasing the signal-to-noise ratio is important in order to investigate the possible link between solar activity and the NA climate.





### 3.3 Composite Analysis

With composite analysis, specific periods with pronounced high and low TSI values are selected from the period AD 850–1849. The response of each atmospheric and oceanic variable analyzed in this study is estimated by subtracting the mean of the atmospheric and oceanic variables in the low TSI period from the respective atmospheric and oceanic variables within high TSI periods. The composite difference maps are given in the fourth column of Figure 2 for the SSTs. Figure 3

shows according results for the SLP and Figure 4 for the geo500.

       The MPI-ESM model indicates that during high TSI there are cooler SSTs away from the European coastline by a maximum of +1.2 K, while the CESM model indicates warmer SSTs at the same regions by a maximum of +0.8 K. The latter result suggests a high diversity in the response, depending on the model used, and the advantage of investigating two profoundly different models to

rule out physically and externally forced robust patterns versus model-dependent and/or internally generated ones. In order to highlight the difference of the individual model responses, we additionally provide in the Appendix the CESM ensemble mean response for all methods and variables (SI Figure S13).

       Considering changes in SLP, a blocking-like pressure pattern with negative centers above

Scandinavia and on the southwest of Nova Scotia, and a positive center (blocking high) that extends from Greenland to the central NA basin is associated with higher TSI values according to composite analysis and the MPI-ESM model (Figure 3). This pattern describes the anomalous atmospheric circulation at sea-level and 500 hPa and resembles the 3rd dominant EOF pattern of geo500 variability in the NA basin (SI, Figure S2-c). The individual CESM ensemble members also show a tendency

toward blocking-like structures, but the blocked regions differ according to the ensemble member under consideration.

### 3.4 Dynamical considerations for periods with reduced and increased TSI

       The results drawn with composite analysis indicate changes in the atmospheric circulation patterns in response to solar forcing. These shifts might happen through a bottom-up mechanism,

linking TSI to the tropospheric and surface circulation. Theory relates shallow diabatic heating to negative pressure anomalies and temperature convection to the east of the heating center (Hoskins and Karoly, 1981). To investigate the relation between diabatic heating and negative pressure anomalies, we correlated the temporal evolution of the 300 hPa geopotential height composite

patterns (TScomp300) to each grid point of the composite surface heat flux patterns (HFLX; including both sensible and latent). The composite maps of the HFLX and the 300 hPa geopotential height (geo300) are shown in the first and second columns of Figure 5, respectively. The TScomp300 was calculated by the projection of the geo300 composite patterns onto the respective geo300 field anomalies. Technically the projection is simply the scalar product of the target anomaly field onto the

respective anomaly time series, i.e. TScomp300 = <geo300|geo300(t)>.

       The results of Figure 5 indicate that there is no clear-cut pattern of HFLX response resulting from composite analysis due to TSI forcing. Regarding the MPI-ESM model, higher values of TSI result in a HFLX surplus over the Nordic Seas and regions of the subtropics, by approximately 16 $W \cdot m^{-2}$. Negative pressure anomalies at 300 hPa are built to the east of such regions as proposed by

Hoskins and Karoly's theory (Figure 5, column 2). The HFLX composite values correlate positively and statistically significantly ($p<0.05$) with the TScomp300 with values of r≈+0.6 (Figure 5, column 3). This positive correlation between the HFLX and the TScomp300 takes place close to the regions where negative pressure anomalies are situated. A high-pressure anomaly is subsequently built over





the central Atlantic, indicating that this wave-like response is most likely the result of diabatic heating.
Statistically significant positive correlations between regions of statistically significant HFLX surplus and negative pressure anomalies are shown by the ensemble members E1 and E4 in the CESM. Model internal variability might be too large for the link between diabatic heating and low-pressure anomalies to be obvious in the CESM ensemble members E2 and E3.

As insolation difference also exerts an influence on zonal winds via changes in the meridional
(mid-to-upper tropospheric) temperature gradient, the according anomalous cyclonic circulation that results from diabatic heating due to TSI forcing should also be reflected by the winds. In this way, we can check for consistency in the simulated results regarding the vertical atmospheric structure. In the SI Figure S11 and Figure S12, we present the pressure-latitude cross-sections of the composite differences for the air temperature (TA) gradient and the mean zonal winds (U), respectively, at
different longitudes (60°W, 0°, and 30°E). The comparison of the respective subplots of the SI Figures S11 and S12 indicates a relationship between the temperature gradient and the zonal wind. Stronger latitudinal warming results in stronger easterlies (negative anomalies) at the high atmospheric levels for a stronger TSI.

## 4 Discussion

### 4.1 Linear Methods

From the linear regression analysis, we find that the maximum SST response is approximately ±0.15 K per 1 SD of TSI, which corresponds to ±0.5 K per W·m$^{-2}$ and is spatially limited only over few NA locations. The results of other studies have also indicated that the magnitude of temperature response to solar irradiance changes is small. A modeling study using the Lean TSI reconstruction
and an energy balance model found that the mean annual linear temperature response of the NH to solar variations for the period AD 1000–1850 is equal to 0.2 K (Crowley, 2000). A study that used the NCAR-CSM model and two solar activity amplitude changes (0.25% and 0.65%) during the period AD 850–1849 found a maximum spatial response of 0.3 K per W·m$^{-2}$. This response was calculated using linear regression on yearly mean surface temperature (Ammann et al., 2007). A more
recent study investigated which forcings were responsible for the near-surface air temperature (SAT) variability of AD 850–1850 on sub-decadal time scales in CMIP5 simulations using the 0.1% TSI scaling factor (Le et al., 2016). These authors identified a relationship between TSI and SAT over the tropics and subtropics but no seasonal or annual relationship over most of the NA basin. Even though these results refer to SAT, they support the relatively small temperature response over the NA basin
that we find.

An important question is whether TSI changes have driven NA SST cooling or warming during the preindustrial era. Regressing grid-point SSTs on the TSI time series showed a general NA warming that is maximum in the Greenland Sea over the period AD 1001–1860 in simulations forced with a 0.24% TSI (Swingedouw et al., 2011). We conducted the same analysis using the MPI-ESM
simulation R1 (SI, Figure S3) and found a temperature pattern that looks similar to the one derived by the study of Swingedouw et al., 2011. The effect of TSI in the NA SSTs in R1 is warming over the NA with the maximum values occurring in the Greenland Sea and being approximately equal to +0.4 K per 1 TSI standard deviation (or +1.3 K per W·m$^{-2}$). However, in the solar-only driven simulations, the regression of TSI shows comparably less SST warming in the Greenland Sea and it
even shows cooling in the central Atlantic (see second column, first row in Figure 2).

A possible reason for the differences between Figure 2 and the SI Figure S3 might not be TSI, as the regression suggests, but that volcanic activity is actually correlated with solar activity due



to the concurrence of TSI minima with periods of increased volcanic activity during the last millennium (SI, Figure S4). Volcanic eruptions are known to produce a boreal-winter warming in some areas of the NH extra tropics due to the intensification of the winter Arctic Oscillation (Shindell et al., 2003). However, this winter warming depends on the strength of the volcanic eruption and also the ability of the models used in realistically representing stratospheric processes in the context of stratospheric warmings due to volcanic aerosols (Zambri and Robock, 2016). The ocean warming in winter caused by volcanoes may partly persist into the summer season and thus counteract the effect of changes in TSI in those regions. The results of the linear regression method are contaminated by the large signal (compared to TSI changes) caused by volcanic outbreaks. Another reason for the differences might be the own model's internal variability or model sensitivity being much higher for weak TSI forcing than for strong forcing (Lovejoy and Varotsos, 2016).

Studies that explore the response of winter climate to TSI forcing have identified a 3-4 year lagged solar response signal (Gray et al., 2016;Gray et al., 2013;Scaife et al., 2013;Yukimoto et al., 2017). Even though it is not in the scope of the current analysis, our preliminary results do not indicate such a response during summer (SI Figures S13-S16). Based on the results of the CESM ensemble, the linear response of the NA SSTs to changes in TSI differs among realizations, being spatially quite heterogeneous, and thus pointing to a prominent influence of models' internal variability. Thus, linear regression might not be the most efficient method in order to isolate the climatic changes due to TSI and especially when comparing model output to the results of empirical studies. Generally, the results based on the linear methods show comparatively weak fingerprints of the TSI forcing on the atmospheric and oceanic variables, which could be within the limits of model intrinsic errors, internal variability, and the simplifications still used by their parameterizations.

## 4.2 MCA vs LIA

Several studies that are based on proxy data have used the MCA-LIA differences to investigate the climate response to TSI changes. Maximum NA SST change of |0.7| K is a relatively large impact, considering that the observed maximum SST warming in the NA in the first decade of the 21st century was +1.2 K relative to the 1951–1980 climatology (Hansen et al., 2006). The results of the R4 realization of the MPI-ESM model and of the ensemble members E1, E2 and E4 of the CESM model agree on the location of the maximum warmer (cooler) SSTs during the LIA (MCA) that is the northeast of Iceland. A recent paleoceanographic reconstruction suggested that the subpolar gyre weakened between the MCA and the LIA transition (Copard et al., 2012), while observations in the NA for the period 1990–2007 showed that a sustained weakening of the NA subpolar gyre allows the increase of the penetration of warm subtropical waters toward the Nordic Seas (Häkkinen et al., 2011). These results could explain the LIA SST warming seen on the Northeast of Iceland in the present study. However, a weakening of the subpolar gyre has instead been connected to anomalously cold conditions in the Nordic Seas during the LIA, but only in full forcing simulations of the past millennium (Moreno-Chamarro et al., 2016b). In the solar-only forced MPI-ESM simulation, no such multi-centennial changes in the subpolar gyre strength are identified (Moreno-Chamarro et al., 2016a).

The oversimplification of using the MCA and the LIA epochs to characterize climate changes during the past millennium assumed in proxy-based studies ignores the fact that these two periods show heterogeneous phases of solar activity, which potentially masks any robust solar-climate relationship. Even though the MCA epoch is supposed to represent predominantly warm climatic background conditions, this does not necessarily imply positive TSI anomalies during the whole period (Figure 1a). Moreover, the predominantly warm phases during the MCA epoch of the last



millennium are not evident by the average summer temperature of the NA basin as simulated by the solar-only forced simulations (SI, Figures S5 and S6). The opposite is not true for the LIA either and

latest reconstructions confirm that the LIA cooling was heterogeneous both in time and space (Neukom et al., 2019). As defined in our study, the mean difference in TSI between the MCA and LIA is of 0.35 W·m$^{-2}$, which corresponds to a 0.03% change in the mean TSI value. Therefore, the climatic response found in terms of near-surface temperature difference, might be too large to be solely explained by changes in solar forcing.

### 4.3 Composite Analysis and Dynamical Considerations


Even though our study regards the summer atmospheric circulation it indicates the existence of blocking highs, similarly reported in studies that investigate winter circulation (García-Herrera and Barriopedro, 2006;Moffa-Sánchez et al., 2014;Saarni et al., 2016;Woollings et al., 2010). For the winter season, higher blocking persistence was found during low solar activity by a model-based

study (Barriopedro et al., 2008), while a proxy-based study supports that solar minima could have induced changes in the stratosphere that favor the development of high-pressure blocking systems located to the south of Greenland (Adolphi et al., 2014). Another modeling study supports that the winter SLP response to solar minima is a North Atlantic Oscillation-like pattern (NAO) (Ineson et al., 2011), while a study based on reanalysis data found that blocking frequency increases along the

coast of Europe when the resulting SLP pattern is a West Atlantic blocking-like pattern (Davini et al., 2012). However, it is unclear whether these results can be extended to summer, as similar studies are lacking.

Identifying the underlying physical mechanisms is key to justifying the way TSI forcing affects climate and ensuring that conclusions are not drawn by statistical coincidences. Our results

suggest that diabatic heating (due to TSI forcing) at the surface of the Atlantic Ocean is responsible or at least favours the development of low-pressure anomalies. This is also in accordance to the linear quasi-geostrophic theory, which supports that low-pressure anomalies form at the east of mid-latitudinal heating (Charney, 1947;Eady, 1949;Wetzel et al., 2017). In the case of the MPI-ESM-P model and the E2, E4 CESM ensemble members we find a wave-train like response at geo500, with

a high pressure anomaly surrounded by low pressure systems. The created southerly flow downstream of the high-pressure system can possibly favour a blocking situation over the central Atlantic during summer solar maxima.

### Conclusions


We investigated the response of the NA summer climate to solar forcing in the preindustrial last millennium. We used a fully forced simulation (volcanic forcing, solar forcing) and solar-only simulations forced with the latest recommendation for a weak scaling of the TSI amplitude in CMIP5. Our study includes ensemble runs in order to identify the effect of model internal variability on the models climatic response to TSI forcing. This analysis is especially relevant for the comparison of

model output to the results of empirical studies, as we investigate the summer season and use methodologies commonly applied in studies looking into the effect of external forcing on past climate. Our results indicate that:

- Linear regression is not a robust method for the isolation of the climatic regional effects of solar forcing. This conclusion was demonstrated by the comparison of a fully forced simulation and a solar-only forced simulation conducted with the MPI-ESM model.



- The SST response found by the method MCA-LIA cannot be attributed to the difference in TSI amplitude between those periods, as these periods do not fall into predominantly high/low phases of solar activity.
- The investigation of individual ensemble members signified that for the low TSI scaling model internal variability is larger than the TSI climatic signal during summer. That might be a reason for the low solar signal in the NA summer climate compared to earlier modeling studies that used larger TSI scaling amplitudes.
- Robust conclusions about regional warming or cooling of the NA SSTs due to changes in TSI forcing cannot be drawn using neither a single model realization nor one ESM.
- Diabatic heating links the TSI surface response to the atmospheric circulation response and induces wave-like pressure anomalies. Such atmospheric conditioning in high TSI periods might favour blocking-like patterns over middle and high latitudes in summer.

**Data Availability**

The CESM-LME output can be downloaded by the Earth System Grid at NCAR.
([https://www.earthsystemgrid.org/dataset/ucar.cgd.ccsm4.CESM_CAM5_LME.html](https://www.earthsystemgrid.org/dataset/ucar.cgd.ccsm4.CESM_CAM5_LME.html))

**Acknowledgements**

The authors thank the CESM modelling group for providing their data. Part of the work was carried out in the framework of the European Initial Marie Curie Training network ARAMACC (Annually resolved Archives of Marine Climate Change). This project has received funding from the European Union's Seventh Framework Programme for research, technological development and demonstration under grant agreement no 604802.

**Author Contributions**

The analysis was performed by M.P. with the consultation of S.W. and E.Z., while E.M.C. conducted the MPI-ESM solar-only forced simulation. M.P. prepared the paper with contributions from all coauthors.

**Competing interests**

The authors declare no competing interests.





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



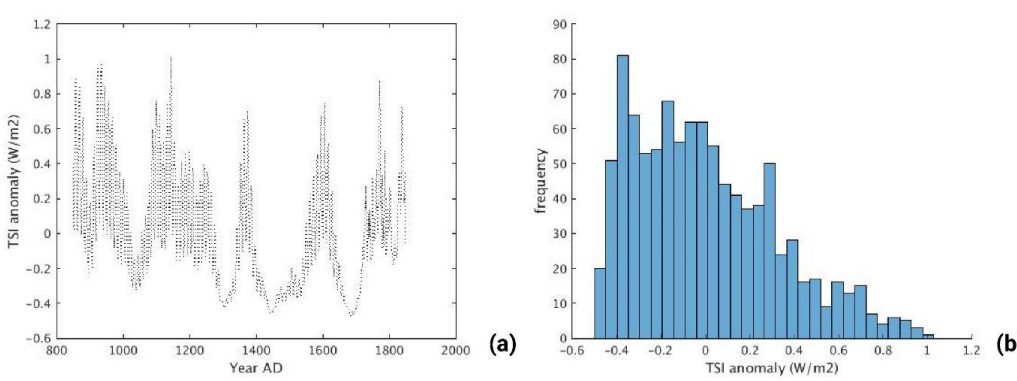

**Figure 1.** Inter-annual TSI anomalies for the period 850–1849 AD presented in a) as a function of time and in b) as a function of frequency.





**Figure 2.** Spatial climatic response patterns of the North Atlantic SSTs due to TSI changes in summer,
calculated with four different methods. The first and second columns respectively show the results of the
correlation and regression between TSI anomalies and NA SSTs for the period 850–1849 AD.  The third and
fourth columns show the results of the methods MCA-LIA and Composite Analysis, respectively. The results
are given in the first row for the realization R4 of the MPI-ESM-P model and in the second, third, fourth and
fifth row for the realizations E1, E2, E3 and E4 of the CESM model, respectively. The hatched areas indicate
statistical significance at the 5% level taking into account the effect of serial correlation. In the case of the
linear regression, the units refer to changes of 1 TSI standard deviation.





**Figure 3.** Spatial climatic response patterns of the North Atlantic SLP due to TSI changes in summer, calculated with four different methods. The first and second columns respectively show the results of the correlation and regression between TSI anomalies and NA SLP for the period 850–1849 AD. The third and fourth columns show the results of the methods MCA-LIA and Composite Analysis, respectively. The results are given in the first row for the realization R4 of the MPI-ESM-P model and in the second, third, fourth and fifth row for the realizations E1, E2, E3 and E4 of the CESM model, respectively. The hatched areas indicate statistical significance at the 5% level taking into account the effect of serial correlation. In the case of the linear regression, the units refer to changes of 1 TSI standard deviation.



**Figure 4.** Spatial climatic response patterns of the North Atlantic geo500 due to TSI changes during the summer, as calculated with four different methods. The first and second columns respectively show the results of the correlation and regression between TSI anomalies and geo500 for the period 850–1849 AD. The third and fourth columns show the results of the methods MCA-LIA and Composite Analysis, respectively. The results are given in the first row for the realization R4 of the MPI-ESM-P model and in the second, third, fourth and fifth row for the realizations E1, E2, E3 and E4 of the CESM model, respectively. The hatched areas indicate statistical significance at the 5% level taking into account the effect of serial correlation. In the case of the linear regression, the units refer to changes of 1 TSI standard deviation.





**Figure 5.** Composite patterns for the North Atlantic HFLX and geo300, Column 1 and Column 2 respectively, due to TSI changes during the summer. The correlation between the TScomp300 and the mean HFLX is given in column 3. The hatched areas indicate significance at the 5% level. The results are given in the first row for the realization R4 of the MPI-ESM-P model and in the second, third, fourth and fifth row for the realizations E1, E2, E3 and E4 of the CESM model, respectively.

860