# Peer review of "Spatial Signature of Solar Forcing over the North Atlantic Summer Climate in the Past Millennium"

_Earth System Dynamics, 2019_

## Referee Comment (RC1) · Anonymous Referee #1 · 22 Nov 2019

The study of Maria Pyrina et al., investigate the spatial signature of solar forcing over the North Atlantic summer climate in the past millennium, which is barely involved in previous works of solar forcing influence. This study proposed three very important questions in their introductions: (1) how the solar forcing interact with the summer surface climate and atmosphere circulation? (2) how is the solar signal transferred to surface? And (3) how do different methods (correlation, regression, composite) condition the understanding of the solar effect? These three questions show the focus and significant of this study. However, there's no conclusive answer in in their following text. Below I describe some concerns/points that the authors may tackle. I think a proper revision will require more time than that is usually given as a major revision and I would suggest a re-submission of a considerably revised manuscript.

1. data and method: The author described the methods firstly and then gave the model simulations. I would suggest show the data (model simulations) first, because many experiments with brief description are involved in the method part, which I have to check and figure out the details in the following "model simulations" section when I read the method section first.

1) There two models used in this study, one is MPI-ESM-P model forced by full external forcings as PMIP3, the other model is CESM-LME forced only by solar forcing. The MPI-ESM-P only has one member (R1) and four ensemble members from CESM-LME (E1, E2, E3, E4). There's no description about what's the difference between these four ensemble members? Different initial conditions? Each ensemble member includes one experiment or more? As the author noticed, the solar signal is weak and maybe concealed by other external forcings (like volcanic eruptions) and internal variability, ensemble mean of solar forced experiment (CESM-LME) provides a possible way to distinguishing the solar signal from internal variability, however, the analyzed results from each ensemble members were shown in this paper, which still includes the internal variability. Is there more than one ensemble members in E1? More details about the ensemble members need to add. I would suggest add a table to shown all the models, experiments, forcing data, ensemble members et al., involved in this study.

2) Is there any difference in the solar forcing data for the two experiments? There's no discussion about this. As described in the "model simulation" section, these two models forced by two different TSI reconstruction data. Figure 1 shows the TSI anomalies, where is this solar forcing data from? Is it one of the solar forcing data for the two experiments?

3) the solar signal at inter-annual and decadal timescale was investigate in this study. How to separate these two timescales? Low-pass filter with 11-year running mean was applied to get the decadal time series. How much the solar signal left in the time series after 11-year running mean? In my experience, the 11-running mean would erase the original solar signal. So please check the 11-year running mean over the TSI firstly

and add this running mean time series in figure 1. And after low-pass filtered, no only the decadal, but also the inter-decadal and longer term will be left. The intern-annual signal is obtained by high-pass filtered? So more information about separating the inter-annual and decadal timescal is needed.

4) composite analysis: this composite was did based on high and low TSI, how many years involved in the high and low TSI respectively? I would suggest add some indications on figure 1 about the high and low TSI. Which timescale this composite result belongs to? The climate variables (SST, SLP, geopotential height) were pass-flitered before composite?

5) A two-tailed Student's t-test was used to test the significance of correlation, however, this involves statistical test of hypothesis that is conducted at individual gird points at nominal significant levels of 5% resulting in fields of test-statistics, where significant values are stippled. it is not apparent that outcomes are "field significant" in the sense of, for example, Livezey and Chen (1983, Monthly Weath. Rev.). This means that the regions in these maps that are contoured as being statistically significant could simply be the result of random error. The fact that these areas tend to be spatially organized, even if not field significant, provides relatively little useful information in single fields (large scale dynamics and thermodynamics naturally impart spatial structure). In addition, the method of significance test for regression and composite is absent in the method section.

2. results and discussions: This paper try to show the responses to solar forcing in SST, SLP, and geopotentail hight (500) through three methods (correlation, regression, composite) on inter-annual and decadal timescales. Due to the unclear method description, the results are not described very clearly and less conclusive results are found for the solar signal and barely physical mechanism described or discussed to help understanding the response patterns. Even though the authors try to show a possible link between TSI, surface heat flux and geopotential height at 300hPa in the composite results, it's still no clear about the possible physical mechanisms involved in

the possible "bottom-up" or "top-down" transport.

1) linear methods and the solar signals: it's hard to expect the correlation or regression analysis can separate the solar signal from internal variability and no surprise the pattern is strongly shaped by the model internal variability, as the solar forcing is very weak at some period (like 1400-1550) and the internal variability is strong in the north Atlantic. So same question as mentioned above, is there many ensemble members in E1? What the internal mode looks like in this region? What the response pattern looks to solar forcing? As the author proposed at the beginning, how dose the solar forcing interact with summer surface climate?

2) composite results: I assume the composite difference maps are the composite in the high TSI years minus low TSI. According to the description in the method section, the high and low TSI are selected in different period, due to no information about the specific years of the high and low TSI, not sure these composite results are the responses on the decadal timescales or on multi-decadal timescales. It seems a blocking-like pressure pattern response to the high TSI forcing in the composite results in both two models, and the block regions depend on the ensemble members. However, no more evidence for this blocking pattern and no figure to show how the blocking regions change according to the number of ensemble members. Further more, the two models using different external forcing data, it's hard to say this blocking pattern is common response to the solar cycle forcing. As this blocking pattern resembles to the 3rd EOF mode of the geo500 variability (inter-annual?), it's necessary make clear which is response pattern to the external forcing or solar forcing, which is the internal mode. Same as the linear methods, in theses composite results, the internal variability at various timescales and the responses to the other external forcing (like volcano) still might be misleading the explanation of the solar signal. The author tried to explain these composite results through the surface heat flux anomaly (bottom-up) and the thermodynamics in troposphere (air temperature latitudinal gradient and zonal wind anomalies), but there's no clear solar-induced heat flux pattern which related to the

blocking response and no clear dynamical mechanism shown in this part. In addition, maybe there's a possible way that the decreased pole-ward air temperature gradient leads to an anomalous easterly wind over the high-latitude in the high solar activity years in these high-top models, but the figures in the supplement file only shown the troposphere and without significance test, it's hard to get some clear information about this "top-down" influence route. So the blocking response in the composite results is not robust in present version, more analysis about the pattern and more discussion about the possible mechanism are needed.

3) Comparison of MCA and LIA: This study also compared the MCA and LIA difference based on two model simulations as these two epochs are related two different solar activity period and significant differences were tested. These analysis may show some light on how the solar forcing influences the surface climate on long-term scale. However, no consistent response found in these two models simulations, as discussed in the paper, the model internal variability may shape the climate change in these two epoch. So it's necessary to separate the external forced component (e.g. solar forced) from the internal variability when checking the response pattern. But, how the "high-frequency" internal variability impacts on the mean state response (long-term) to the external forcings? Some analysis or discussion about this part might be help to understand the results shown in this study.

This study using two high-top model simulations to investigate the solar signal in the north Atlantic summer climate in the past millennium, it's a new work at some extent because most previous works are focus on the winter season. But no consistent (conclusive) spatial signature of solar forcing is found in this study, no answers to the supposed questions at the beginning, the whole story is not clear as the the responses are mixed up at different timescales and the internal variability included in every individual simulations.

---

## Referee Comment (RC2) · Anonymous Referee #2 · 10 Dec 2019

The authors analyze solar signals in simulations of the last preindustrial millennium with two different Earth system models. Their focus is on spatial patterns of surface climate responses during summer in the North Atlantic region. Three different analysis techniques are used, regression and correlation of interannual variability, comparison of the extended periods of the Medieval Climate Anomaly (MCA) and the Little Ice Age (LIA), and comparison of composites built from years for relatively high and low solar activity. The main conclusion, in my eyes, is that in the model simulations one can identify very little robust signals. Analyzed signals in, e.g., sea surface temperatures (SST) differ between the models, between different ensemble members of simulations with the same model, and between different analysis techniques. I don't think this result is surprising as the applied solar forcing is small, and there has, so far, no mechanism

been suggested for any amplification of solar signals for this space and time as it has been for example for the wintertime northern hemisphere where stratospheric signals may propagate downward through wave coupling. The question if solar signals can be simulated in NH summer can be of interest even if the answer in this case is rather a No, because it might help to understand signals in proxies that may be largely influenced by summertime conditions. I find it even useful to show that apparent signals in 1000-year long simulations may be spurious. However, I think the quality of the presentation and interpretation of the analysis is not sufficient to warrant publication, and I would suggest to reject the publication of the current manuscript. In the following I will provide reasons for this, mainly by discussing the conclusions provided by the authors, followed by a few more general comments. I won't provide a detailed line by line analysis at this stage of the review as I think a considerably reworked resubmission is necessary anyhow.

Discussion of conclusions: "Linear regression is not a robust method for the isolation of the climatic regional effects of solar forcing. This conclusion was demonstrated by the comparison of a fully forced simulation and a solar-only forced simulation conducted with the MPI-ESM model." As much of the manuscript this statement is not only problematic concerning its content but also concerning the often very approximate wording. I guess what the authors mean is not that the method is not robust but that it provides no robust results. Even then the statement is untrue. I don't dispute that the linear assumption can be very problematic. However, many solar signals have been robustly identified using regression techniques. Here, the authors very likely refer to the identification for the specific season and region they have studied. It's true that if at all very few patterns can be identified robustly in the two MPI-ESM simulations. If this is considered the first main conclusion of the paper, I'm wondering why the authors hide one of the figures (S3) supporting this statement in the supplementary material and use a different color scale than for the corresponding Fig. 2. Furthermore, from Figs. 2-4 it seems to me that the issue is not different for the individual ensemble members of the CESM and e.g. for the analysis of composites. If one of the conclusions should be that composite analysis is more appropriate than correlation analysis, I think it is necessary

to substantiate this qualitatively.

"The SST response found by the method MCA-LIA cannot be attributed to the difference in TSI amplitude between those periods, as these periods do not fall into predominantly high/low phases of solar activity." I'm not sure what the authors mean when they speak of the "SST response found by the method MCA-LIA". Looking at Fig. 2, it again seems to me that there is not much of a robust signal, not different to correlation and regression analysis. If the goal is to identify solar signals then why chose periods from which one knows that they are not appropriate. Furthermore, this statement exemplifies again the careless usage of language. The authors very often speak of responses or signals when it is not at all clear if the analyzed anomalies indeed represent responses or signals.

"Robust conclusions about regional warming or cooling of the NA SSTs due to changes in TSI forcing cannot be drawn using neither a single model realization nor one ESM." What would the authors like to identify robustly? The solar signal in the real world? I would argue that this can't be done with model simulations at all. Even if models showed robust responses, they might all suffer from the same mistakes, e.g. in the imposed forcing. I guess what the authors want to say is that the simulations analyzed here show virtually no robust responses. Analyzed anomalies differ between models and even between different ensemble members simulated with the same model.

"Diabatic heating links the TSI surface response to the atmospheric circulation response and induces wave-like pressure anomalies. Such atmospheric conditioning in high TSI periods might favour blocking-like patterns over middle and high latitudes in summer." Here it seems like the authors argue they have identified a robust signal: a tendency for blocking at higher solar irradiance. It would be nice to quantify this so that one doesn't have to formulate as carefully as the authors do ("might favour"). I have to admit that I'm not a blocking expert. It may be difficult to analyze blocking from model output which I guess is only available at coarse temporal resolution. But from visual analysis of the figures alone I'm not sure about the blocking statement. Indeed, the SLP

anomaly from the MPI-ESM composite analysis suggests more zonal inhomogeneity for higher TSI. But I'm less certain about the other simulations and hence robustness. Furthermore, I have difficulties to follow the reasoning concerning the diabatic heating and subsequent circulation changes. It is of course true that thermodynamic changes would entail changes in dynamics. But why identify diabatic heating from turbulent energy fluxes alone? What about radiative fluxes? And how does this analysis help me to identify if anomaly patterns may be of solar origin or due to internal variability?

A few more general comments: As mentioned with respect to the conclusion, I think the use of language is in most parts of the paper inappropriate for a scientific publication. There are almost no typos or errors of grammar, but formulations are often inexact. However, it is not only language, but also the descriptions of the simulations (e.g. how varies the SSI in the CESM simulations) and the figure captions (e.g. unit of column 2 of Figs. 2-4) are partly inexact. Given the experience of several of the co-authors this issue should be solvable with some more effort. This should not be a task for the reviewers.

One issue I have with the analysis of NA summer is that I don't know how the model performs for other parameters for which more information on solar signals exists. E.g., CMIP5 models have been shown to relatively robustly simulate global mean near surface temperature responses to 11-year solar variability, the maximum occurring about 2 years after solar cycle maximum. Do the models analyzed here show a similar behavior? If this were not the case, why should one bother to analyze NA summer. Similarly it would be interesting to learn about the models' responses for example in the tropics or NH winter. And there should be some information on the simulation of the NH summer climatological state. All this could go into the supplementary material.

Furthermore, it is necessary to formulate hypotheses about the expected responses analyzed with the three different methods. These responses might be very different, e.g. due to the different time scales involved. As alluded to above, I don't think that an analysis of interannual variability without testing time lags is sufficient. For the

composite analysis it would be important to discuss the temporal distribution of the years contributing to the composites. Are they more or less alternating with the 11-year cycle so that one could interpret differences to the regression technique as method-related, or do they often sample longer periods with lower or higher solar irradiance?

As said initially I do think that the analysis attempted in this paper can be of value, so I would like to encourage the authors to invest more and resubmit the paper.

---

## Author Comment (AC1) · 22 Jan 2020

First of all the authors would like to thank Referee 1 for agreeing to review this manuscript. We appreciate the effort and the comments, which, we are sure, will significantly improve the current manuscript.

The motivation for the present study was that the influence of solar forcing over the past millennium has been detected in marine proxies for the Sea Surface Temperature (SST) of the North Atlantic (NA) region in centennial and multi-decadal time scales (Jiang et al., 2005; Moffa-Sánchez et al., 2014; Sejrup et al., 2010). Therefore, in the current version of the manuscript, we wanted to test whether 1000-year long solar only forced climate simulations also show this influence and at which time scales (inter-

annual, decadal) can be detected. It is important to use solar only forced simulations especially because of the low TSI scaling used in our study, which is the current standard value used for solar forcing in many CMIP5 simulations (Kopp and Lean, 2011; Prša et al., 2016). The low TSI scaling might most likely be the reason why the solar signal does not appear robust and difficult to discriminate from internal variability. Our analysis does, therefore, provide a scientific contribution since it shows that the interpretation of the proxy signals in the aforementioned papers is not totally consistent with the modelling results.

In a revised version of the manuscript we will refocus the manuscript and we will robustly show whether we identified a solar forced response in the 1000-year long simulations, and discuss the implications that this might have for comparisons between models and proxy data. We will therefore implement an additional chapter specifically dedicated on the discrimination between internal and solar forced signals. This result is important in order to support or put into question the interpretation of the solar signal in proxy records. Our null-hypothesis will be that the CESM simulations do not show any response to changes in solar activity. The null-hypothesis will be tested through point wise correlations between the individual CESM ensemble members. The null-hypothesis can eventually be rejected for those regions (taking care of global significance) showing correlations exceeding the 5% significance level. The CESM solar only forced ensemble members are forced with the same TSI forcing (they differ slightly in the initial conditions), therefore regions with significant values of temporal correlations among the members will be identified as regions that respond to the common forcing. In the case that we identify a robust solar signal, then for determining the time scales of the signal we will additionally use cross-wavelet analysis of the climatic variable and solar forcing, for the NA sub-regions that are indicated by the CESM ensembles as regions with commonly forced signal. To test the significance of the periodicities of the signal, we will investigate the existence of similar periodicities in synthetic time series with prescribed statistical properties.

The variables that will be analyzed will be the SSTs (motivated by the aforementioned proxy studies) and the turbulent heat fluxes (key parameters for the interactions between ocean–atmosphere and the transmission of a possible solar signal). We will also investigate variables related to atmospheric dynamics that were shown to respond to solar forcing by previous studies for winter circulation (i.e. sea level pressure, zonal wind; Thieblemont et al., 2014). All of these variables were already analyzed in this manuscript but not appropriately presented, according to both of the Reviewers' comments. The main conclusion is that the solar signals are not robust in these variables either, and therefore no robust physical mechanisms during summer can be identified. We believe that a revision of this manuscript will still be inside the time guidelines of a major revision. In the following, we provide a point-by-point answer to the Reviewer's comments.

Answers to the reviewer's comments on 1. Data and Methods:

1) We will follow the reviewer's recommendation and include a table with the models, experiments, forcing data and ensemble members involved in this study. Regarding internal variability, we agree with the reviewer that the ensemble mean of the solar forced CESM experiment provides a possible way to distinguish the solar signal from internal variability. Nevertheless, it might be that a large number of the ensemble members is needed in order to do so and that this number depends on the problem (Maher et al., 2018) and region (Bengtsson et al., 2019). Moreover, not all ensemble members should be equally represented in this mean (Wanders and Wood 2016). The CESM solar only forced ensemble provides four ensemble members, the ones analyzed in the current study. In this study, we show the response patterns estimated from the individual ensemble members and how these patterns might vary due to the effect of the different realizations of internal variability that each of the members contains. In this way, we can better compare model results to results from empirical studies, as this approach is more representative regarding the uncertainty in the forced response arising from internal climate variability. However, as stated in the line 321, we have provided

in the Appendix of the current manuscript the CESM ensemble mean response for all methods and variables (SI Figure S13).

2) All the experiments presented use the same TSI forcing, which is the Vieira et al., (2011) total solar irradiance reconstruction (referenced in lines 227, 232). In the revised version of the manuscript, this will be clearly shown in a table and it will referenced in Figure 1.

3) Regarding the inter-annual time scale there is no filtering of the data. The low-pass filter was applied on the variables (SST, SLP, geo500) in order to isolate the signals with a frequency lower than the selected cutoff frequency (fmax=1/11 in our case). Therefore, variability was suppressed for time scales shorter than 11 years. The design of the filter is cited in the manuscript (line 156). There was no filtering applied on the TSI data because the physical resolution of the reconstructed TSI record is already decadal (Vieira et al., 2011), due to the resolution of the ice-core records. Additionally, there was a mistake on the text referring to an "eleven-year running mean filter", when actually it should have been referred to as "eleven-year low pass filter", as stated in the next sentence (line 156). The reviewer is correct that after low-pass filtering, not only the decadal, but also the inter-decadal and longer time scale variability will be retained, but the effect of these time scales is not expected to affect the results of this study on decadal time scales.

4) Regarding composite analysis we agree with the reviewer about stating the number of years involved in the high and low TSI respectively and adding some indications on Figure 1 about the high and low TSI values. The composite results are given only for inter-annual time scales, as the results of this method will not be interpretable after low pass filtering the data. Therefore, the climate variables (SST, SLP, geopotential height) were not low pass filtered prior to composite analysis. This will be also specified in the new manuscript version.

5) In the last paragraph of the section 2.1 Methods, we mention the significance tests

used for all methods: "A two-sided Student's t-test that accounts for the effect of se-rial correlation was applied on the results of the linear methods. A mean difference two-sided test was also conducted to evaluate the statistical significance of the results according to the methods MCA minus LIA and composite analysis." We agree though that we do not clarify that the significance tests were indeed performed for each indi-vidual grid point. Regarding the field significance of our results, we will apply a field significance test for identifying whether the basic conclusions of our results are robust. This can be achieved through Monte Carlo temporal re-shuffling in order to retain the spatial auto-correlation of the climate fields.

Answers for the reviewer's comments on 2. Results and Discussion:

1) The E1, E2, E3, E4 experiments are the ensemble members of the: solar only forced experiment with CESM (lines 160-162). We will make clear in the new version of the manuscript that the difference in the CESM ensemble members is only the slightly different initial conditions. As described in the discussion, there are studies that use regression and correlation to show the relationship between solar forcing and climate. Here, we tried to show that internal variability might be stronger than the signal itself, which would have important implications for the comparison of model output to the results of empirical studies and the interpretation of the identified proxy signals. The individual CESM ensemble member response patterns to solar forcing are shown in Figures 1-4 and the ensemble mean response pattern in the Appendix Figure S13, as stated in the manuscript (line 321). The illustration of the signal derived from individual members is useful to provide a measure of the impact of internal variability. Regarding the interaction of solar forcing with summer surface climate, there is a separate section (3.4) that includes only the results drawn using composite analysis.

2) The results of composite analysis regard high TSI years minus low TSI years, as stated in line 313. We believe that this reviewer's comment will be addressed in the new version of the manuscript where the data and methods sections will be clearer for the reader.

3) As the reviewer suggests, in the new version of the manuscript we will further discuss how the "high-frequency" internal variability blurs the estimation of the response to the external forcings.

Bengtsson, Lennart, and Kevin I. Hodges. "Can an ensemble climate simulation be used to separate climate change signals from internal unforced variability?." Climate Dynamics 52.5-6 (2019): 3553-3573.

Jiang, H., Eiríksson, J., Schulz, M., Knudsen, K. L., & Seidenkrantz, M. S. (2005). Evidence for solar forcing of sea-surface temperature on the North Icelandic Shelf during the late Holocene. Geology, 33(1), 73-76.

Kopp, G., and Lean, J. L.: A new, lower value of total solar irradiance: Evidence and climate significance, Geophysical Research Letters, 38, 2011.

Maher, Nicola, et al. "ENSO change in climate projections: forced response or internal variability?." Geophysical Research Letters 45.20 (2018): 11-390.

Moffa-Sánchez, P., Born, A., Hall, I. R., Thornalley, D. J., & Barker, S. (2014). Solar forcing of North Atlantic surface temperature and salinity over the past millennium. Nature Geoscience, 7(4), 275-278.

Prša, A., Harmanec, P., Torres, G., Mamajek, E., Asplund, M., Capitaine, N., Christensen-Dalsgaard, J., Depagne, É., Haberreiter, M., and Hekker, S.: Nominal values for selected solar and planetary quantities: IAU 2015 Resolution B3, The Astronomical Journal, 152, 41, 2016.

Sejrup, H. P., Haflidason, H., & Andrews, J. T. (2011). A Holocene North Atlantic SST record and regional climate variability. Quaternary Science Reviews, 30(21-22), 3181-3195.

Thiéblemont, R., Matthes, K., Omrani, N. E., Kodera, K., & Hansen, F. (2015). Solar forcing synchronizes decadal North Atlantic climate variability. Nature communications, 6, 8268.

Wanders, Niko, and Eric F. Wood. "Improved sub-seasonal meteorological forecast skill using weighted multi-model ensemble simulations." Environmental Research Letters 11.9 (2016): 094007.

---

## Author Comment (AC2) · 22 Jan 2020

The authors would like to thank Referee 2 for agreeing to review the manuscript, as well as for the thorough revision and suggestions.

The authors agree with the reviewer that the low TSI scaling used in the current study is most likely the reason for the signal not appearing robust, but that is the current standard value used for solar forcing in many CMIP5 simulations (Kopp and Lean, 2011; Prša et al., 2016). As the reviewer pointed out, the question whether a solar signal can be detected in NH summer is of interest even if the answer is negative. This result is important in order to support or put into question the interpretation of the solar signal, which as it has been claimed is reflected in proxy records (Jiang et al., 2005;

[Figure]

Moffa-Sánchez et al., 2014; Sejrup et al., 2010).

In the following, we provide a point-by-point answer to the reviewer's comments. Specifically:

1. In the revised version of the manuscript, we will substantiate qualitatively the differences in the methods' results. We will refocus the manuscript, robustly show whether we identify a clear solar forced response in the 1000-year long simulations, and discuss the implications that this might have for comparisons between models and proxy data. We will therefore implement an additional chapter specifically dedicated on the discrimination between internal and solar forced signals. Our null-hypothesis will be that the CESM simulations do not show any response to changes in solar activity. The null-hypothesis will be tested through point wise correlations between the individual CESM ensemble members. The null-hypothesis can eventually be rejected for those regions (taking care of global significance) showing correlations exceeding the 5% significance level. Significant temporal correlations among the members will identify regions that respond to the common forcing. In the case that we identify a robust solar signal, then for determining the time scales of the signal we will additionally use cross-wavelet analysis of the climatic variable and solar forcing, for the NA sub-regions that are indicated by the CESM ensembles as regions with commonly forced signal. To test the significance of the periodicities of the signal, we will investigate the existence of similar periodicities in synthetic time series with prescribed statistical properties.

2. The periods Medieval Climate Anomaly and Little Ice Age were chosen in order to show that it is problematic to identify the solar signal on SSTs during periods which, even though might turn out not to be appropriate for this purpose, they were indeed used by other studies for the investigation of surface response to solar forcing (i.e. Mann et al., 2009; Deng et al., 2017). In the revised version of the manuscript for signal identification, we will use linear methods and composite analysis, but we will exclude the method MCA-LIA. Moreover, we will use a consistent terminology in the updated version and use the term "response" instead of "signal" if changes in solar
activity are evident in climatic variables.

3. The reviewer is correct that we mean that the simulations analyzed here show virtually no robust responses. We will rephrase and use more carefully the language in the revised version.

4. We analyzed diabatic heating from turbulent energy fluxes because the heat extracted from the ocean by the atmosphere is related to the air–sea turbulent fluxes and therefore implies a decrease or increase, respectively, in SST. In the revised version of the manuscript, we will address the robustness of the atmospheric response using the point wise correlations among the CESM ensemble members. As stated previously, the null-hypothesis will be that the CESM simulations do not show any response to changes in solar activity.

5. In a revised version we will cite other studies using the same models or calculate how the ESMs that we used simulate: a) global mean near surface temperature responses to 11-year solar variability b) models' responses in the tropics or NH winter and add supplementary information on the simulation of the NH summer climatological state.

6. Regarding the formulation of hypotheses about the expected responses analyzed with the three different methods, we have explicitly formulated questions (lines 64-65, 359-361). We can extend those questions to expected outcomes (i.e. a direct signal manifested in increased sea-surface or land-temperatures during periods with increased TSI and vice versa) or to other variables such as SLP or geopotential height. However, the response of such variables might be more difficult to hypothesize, as to date no clear hypotheses exist for the summer circulation caused by changes in TSI over the North Atlantic region. If anything, one might use the hypotheses formulated in the context of changes in circulation during winter and test whether similar response can be found for summer.

7. Regarding a time lagged analysis, as stated in the line 411, we have indeed provided with preliminary results. These results do not indicate a lagged response during

summer (SI Figures S14-S16). They indicate that introducing a lag does not change the qualitative nature of our results. In a revised version, we will elaborate more on the results of this analysis. Moreover, the lead-lag issue would implicitly be resolved by the point-by-point correlation analysis among the CESM ensemble members. Assuming that the physical mechanisms leading to different lags in the different regions of the North Atlantic are the same for the different realizations, then the results of the correlations will include spatially different lag structures in the same plot.

8. Regarding composite analysis, we will also add a discussion on the temporal distribution of the years contributing to the composites.

Deng, Wenfeng, et al. "A comparison of the climates of the Medieval Climate Anomaly, Little Ice Age, and Current Warm Period reconstructed using coral records from the northern South China Sea." Journal of Geophysical Research: Oceans 122.1 (2017): 264-275.

Jiang, H., Eiríksson, J., Schulz, M., Knudsen, K. L., & Seidenkrantz, M. S. (2005). Evidence for solar forcing of sea-surface temperature on the North Icelandic Shelf during the late Holocene. Geology, 33(1), 73-76.

Kopp, G., and Lean, J. L.: A new, lower value of total solar irradiance: Evidence and climate significance, Geophysical Research Letters, 38, 2011.

Mann, Michael E., et al. "Global signatures and dynamical origins of the Little Ice Age and Medieval Climate Anomaly." Science 326.5957 (2009): 1256-1260.

Moffa-Sánchez, P., Born, A., Hall, I. R., Thornalley, D. J., & Barker, S. (2014). Solar forcing of North Atlantic surface temperature and salinity over the past millennium. Nature Geoscience, 7(4), 275-278.

Prša, A., Harmanec, P., Torres, G., Mamajek, E., Asplund, M., Capitaine, N., Christensen-Dalsgaard, J., Depagne, É., Haberreiter, M., and Hekker, S.: Nominal values for selected solar and planetary quantities: IAU 2015 Resolution B3, The Astro-

nomical Journal, 152, 41, 2016.

Sejrup, H. P., Haflidason, H., & Andrews, J. T. (2011). A Holocene North Atlantic SST record and regional climate variability. Quaternary Science Reviews, 30(21-22), 3181-3195.